# The Organogermanium Compound THGP Suppresses Melanin Synthesis via Complex Formation with L-DOPA on Mushroom Tyrosinase and in B16 4A5 Melanoma Cells

**DOI:** 10.3390/ijms20194785

**Published:** 2019-09-26

**Authors:** Junya Azumi, Tomoya Takeda, Yasuhiro Shimada, Hisashi Aso, Takashi Nakamura

**Affiliations:** 1Asai Germanium Research Institute Co., Ltd. Suzuranoka 3-131, Hakodate, Hokkaido 042-0958, Japan; j.azumi@asai-ge.co.jp (J.A.); tomo.t621@asai-ge.co.jp (T.T.); y.shimada@asai-ge.co.jp (Y.S.); 2Cellular Biology Laboratory, Graduate School of Agricultural Science, Tohoku University, Miyagi, Sendai 980-8572, Japan; asosan@tohoku.ac.jp

**Keywords:** organogermanium, THGP, melanogenesis

## Abstract

The organogermanium compound 3-(trihydroxygermyl)propanoic acid (THGP) has various biological activities. We previously reported that THGP forms a complex with *cis*-diol structures. L-3,4-Dihydroxyphenylalanine (L-DOPA), a precursor of melanin, contains a *cis*-diol structure in its catechol skeleton, and excessive melanin production causes skin darkening and staining. Thus, the cosmetic field is investigating substances that suppress melanin production. In this study, we investigated whether THGP inhibits melanin synthesis via the formation of a complex with L-DOPA using mushroom tyrosinase and B16 4A5 melanoma cells. The ability of THGP to interact with L-DOPA was analyzed by ^1^H-NMR, and the influence of THGP and/or kojic acid on melanin synthesis was investigated. We also examined the effect of THGP on cytotoxicity, tyrosinase activity, and gene expression and found that THGP interacted with L-DOPA, a precursor of melanin with a *cis*-diol structure. The results also showed that THGP inhibited melanin synthesis, exerted a synergistic effect with kojic acid, and did not affect tyrosinase activity or gene expression. These results suggest that THGP is a useful substrate that functions as an inhibitor of melanogenesis and that its effect is enhanced by combination with kojic acid.

## 1. Introduction

Inorganic germanium is used in semiconductors and various optical applications [1]. The organogermanium compound poly-*trans*-[(2-carboxyethyl) germasesquioxane] (Ge-132; repagermanium) was first synthesized by Oikawa et al. in 1967, and its structure was reported in 1976 [2,3]. A hydrolysate of Ge-132,3-(trihydroxygermyl) propanoic acid (THGP) exerts various physiological effects, including anti-inflammatory, pain relief, immunostimulatory, and anticancer effects [4,5,6,7,8]. In addition, various animal experiments and clinical trials have shown that THGP is safe [9,10,11,12], and Ge-132 is currently used as a supplement in health foods and cosmetics in Japan, China, South Korea, and the United States.

However, the mechanism underlying the physiological effects remains unclear. In recent years, studies have shown that THGP interacts with a *cis*-diol structure and suggested that this interaction is involved in the physiological effects of THGP [13,14]. The identified biological substances with a *cis*-diol structure include catecholamine, ATP, and adenosine. L-3,4-Dihydroxyphenylalanine (L-DOPA) is another substance with a *cis*-diol structure. In this study, we performed a nuclear magnetic resonance (NMR) analysis to reveal that THGP forms a complex with L-DOPA.

Melanin is an important factor in the determination of human skin and hair color [15,16]. Exposure of the skin to sunlight, including UV-B, induces keratinocytes to release α-melanocyte-stimulating hormone (α-MSH) [17]. Melanocytes produce melanin in response to this stimulus and transfer melanin to keratinocytes. Melanin then accumulates around the nuclei of keratinocytes and prevents UV-B-induced DNA damage and carcinogenesis [18,19]. However, the excessive production of melanin causes dermatological problems, such as the formation of freckles, solar lentigo (age spots), and melasma [20]. Thus, substances that inhibit the production of melanin have been investigated [21].

The synthesis of melanin consists of several processes [22]. The reaction starts with L-tyrosine, and the tyrosinase enzyme mediates the formation of L-DOPA and dopaquinone. Moreover, dopaquinone is metabolized into melanin via dopachrome, and tyrosinase-related protein 1 (*TRP-1*) and tyrosinase-related protein 2 (*TRP-2*) mediate this process [23,24]. Furthermore, microphthalmia-associated transcription factor (*MITF*) promotes the expression of *tyrosinase*, *TRP-1*, and *TRP-2* [25]. In contrast, kojic acid, arbutin, and similar substances are reportedly melanogenic inhibitors, and many of these substances inhibit the activities of these enzymes [26,27,28,29,30]. THGP does not inhibit enzyme activity and acts on L-DOPA, which is a substrate in melanin synthesis. Thus, the mechanism of THGP differs from that of the other abovementioned inhibitory substances. Therefore, a synergistic effect can be expected from the combination of THGP with kojic acid, which inhibits enzyme activity.

In this study, we investigated the formation of the THGP and-L-DOPA complex and the inhibition of melanin synthesis and its mechanism via complex formation. Furthermore, the synergistic suppressive effect obtained from the combined use of THGP and kojic acid on melanin production was examined.

## 2. Results

### 2.1. THGP and L-DOPA Form a Complex via a Cis-Diol Structure

We evaluated the ability of L-DOPA to form a complex with THGP or 3-trimethylgermylpropanoic (TMGP), which contains a methyl group in place of the hydroxyl group on THGP, by ^1^H-NMR (see Figure 1A). The signals at 1.5 and 2.5 ppm originated from THGP, and the L-DOPA signals are shown at 3.1, 3.9, and 6.9 ppm. The mixture of THGP and L-DOPA produced a spectrum that differed from the characteristic spectra of THGP and L-DOPA alone (Figure 1B). The mixture of these substances yielded novel signals at 1.8 and 2.6 ppm, which were not found on the L-DOPA or THGP spectra and were thus derived from the complexed compounds. However, the mixture of THGP and L-DOPA also showed the individual spectra of THGP and L-DOPA, which suggests that THGP does not form a complete complex with L-DOPA. In contrast, after the mixing of TMGP (signals at 0.1, 1.0, and 2.2 ppm) with L-DOPA (Figure 1C), only the signals from the individual compounds were observed. In summary, the data suggest that THGP forms a complex with L-DOPA through the hydroxyl group of THGP (Figure 1D).

### 2.2. THGP Inhibits Melanin Synthesis through a Reaction Mediated by Mushroom-Derived Tyrosinase

We confirmed that the mixing of L-DOPA and mushroom tyrosinase synthesized melanin. The resulting melanin solutions without added organogermanium (left) or with added THGP (center) or TMGP (right) are displayed in Figure 2A. The melanin content of the control (without added organogermanium) is shown as a percentage in Figure 2B and compared with the control condition. THGP significantly suppressed melanin synthesis (*p* < 0.01) at a concentration of at least 1000 μM (Figure 2A,B). THGP also inhibited melanin synthesis in a dose-dependent manner (93% inhibition with a concentration of 1000 μM to 71% inhibition with a concentration of 5000 μM), whereas TMGP did not inhibit melanin synthesis. This result suggests that the hydroxyl group on THGP is important for melanin synthesis, and THGP might thus inhibit the synthesis of melanin by complexing with L-DOPA via the hydroxyl group of THGP. Kojic acid also inhibited melanin synthesis in a dose-dependent manner (Figure 2C).

The synergistic inhibitory effects between THGP and kojic acid are shown in Figure 2D. The theoretical value for the combination of THGP and kojic acid was considered the standard (with a value of 1), and the synergistic effect of THGP was calculated in comparison to this standard. For example, 5000 μM THGP alone resulted in a melanin content of 67.8% compared with the control, and 1000 μM kojic acid alone yielded an inhibition rate of 51.9%. The theoretical value was found to equal 35.2%, which corresponds to the multiplication of 67.8% by 51.9%. However, because the experimentally measured inhibition rate was 12%, the synergistic effect was 2.93-fold higher than the theoretical value. This synergistic effect was due to the interaction of THGP with the substrate and the interaction of kojic acid with the enzyme.

### 2.3. THGP Inhibits Melanin Production by the B16 4A5 Melanoma Cell Line

The effect of THGP or kojic acid on B16 4A5 melanoma cell proliferation was evaluated through the MTS assay. The results showed that 5000 μM THGP did not affect cell proliferation, whereas 5000 μM kojic acid showed cytotoxicity (80% proliferation; Figure 3). This result suggests that THGP is less cytotoxic and safer than kojic acid.

We examined the effect of THGP or kojic acid on melanin synthesis by B16 4A5 melanoma cells. The melanin contents in the extracellular solution are shown in Figure 4A, and the intracellular melanin contents are shown in Figure 4B. The values in both figures are displayed as percentages relative to the melanin content in the untreated control cells. The amount of extracellular melanin after being treated with 50, 500, and 5000 μM THGP decreased to approximately 60%, 40%, and 30% of the control levels, respectively (Figure 4A). In addition, 500 μM THGP suppressed melanin production by 62%, whereas 500 μM kojic acid exerted an inhibitory effect of 55%. These results suggest that the inhibitory effect of THGP on melanin production was as high as that of kojic acid at the same concentration (500 μM). However, treatment with 5000 μM THGP increased the intracellular melanin content by approximately 1.8-fold compared with the control (Figure 4B). This figure does not show the ratio of the extracellular to intracellular melanin contents, but the extracellular melanin content was approximately 5-fold higher than the intracellular melanin content. This result suggests that the total amount of intracellular and extracellular melanin was suppressed by THGP. Cells treated with 6 mM THGP for 48 h were observed under an optical microscope, and their morphologies are shown in Figure 4C. The cellular accumulation of melanin clearly increased after THGP treatment. These results suggest that the THGP and-L-DOPA complex inhibited not only the synthesis but also the release of melanin. Moreover, the extracellular melanin content of cells treated with the combination of THGP and kojic acid is shown in Figure 4D. The results indicate that the extracellular melanin level was synergistically suppressed by the combination of THGP and kojic acid. In particular, melanin production was suppressed by approximately 90% by the combination of 500 μM THGP with 50 μM kojic acid.

### 2.4. THGP Does Not Affect Tyrosinase Activity or the Expression of Melanin Synthesis-Related Genes

The influence of THGP treatment on the tyrosinase activity of B16 4A5 melanoma cells was examined. The activities observed after different THGP treatments are shown in Figure 5A, and the mRNA expression levels of genes related to melanin production in B16 4A5 cells are shown in Figure 5B. The results show that THGP did not affect tyrosinase activity (Figure 5A), and the expression levels of *tyrosinase*, *Trp-1*, *Trp-2*, and *MITF*, which are the melanin production-related genes, were also not impacted by treatment with THGP alone or in combination with L-DOPA (Figure 5B,C). These results suggest that THGP does not influence the tyrosinase activity or gene expression to inhibit the synthesis of melanin by forming a complex with L-DOPA.

### 2.5. THGP Suppresses Melanin Production in α-MSH- or L-DOPA-Induced Melanogenesis

Because melanogenesis is stimulated by α-MSH, we evaluated the influence of THGP treatment in the presence of α-MSH in the culture media on melanin synthesis and the extracellular melanin content, and the results are shown in Figure 6A. The amount of extracellular melanin increased after treatment with α-MSH (from approximately 70% to 100%, i.e., 1.4-fold increase), and treatment with THGP at concentrations above 50 μM reduced the extracellular melanin levels compared with those obtained with α-MSH treatment (Figure 6B). In particular, the addition of THGP at high concentrations (500 and 5000 μM) completely suppressed melanogenesis. The extracellular melanin content after the addition of the L-DOPA substrate is shown in Figure 6B. Compared with the control condition, the addition of L-DOPA markedly accelerated melanogenesis in melanoma cells by approximately 10-fold. In addition, the cells treated with L-DOPA, THGP, and kojic acid exhibited decreased melanin contents compared with the control cells. THGP (at concentrations of 5, 50, 500, and 5000 μM) inhibited melanogenesis in a dose-dependent manner (to levels that were 90%, 85%, 60%, and less than 10% of the control levels, respectively). At most high concentrations (5000 μM), THGP decreased the melanin level to the same level found in the untreated group (without L-DOPA), but the effect of kojic acid treatment was lower (approximately 90% to 85%) than that obtained with THGP treatment (Figure 6B). This result might reflect the different interactions of THGP and kojic acid: THGP complexes with L-DOPA, whereas kojic acid inhibits tyrosinase activity.

## 3. Discussion

In this study, we evaluated the ability of THGP to suppress melanogenesis. We first attempted to confirm the formation of a complex between THGP and L-DOPA as an early substrate in melanin synthesis. In 2015, the ability to complex with a substance containing a *cis*-diol structure was reported to be a chemical property of THGP [31]. Moreover, a previous study provided the first confirmation that THGP functions physiologically in vitro via the formation of a complex with *cis*-diol compounds [13]. Specifically, THGP suppresses the influx of Ca^2+^ by interacting with adrenaline and ATP, which have *cis*-diol structures. Other studies have demonstrated that THGP interacts with adenosine, D-glucose, and D-fructose [14,32]. These results suggest that the physiological activities of THGP are mediated by binding to *cis*-diol structures. Therefore, we considered whether THGP could form a complex with L-DOPA, which would affect melanogenesis, and our ^1^H-NMR analysis clarified that THGP can also form a complex with L-DOPA, which is a precursor substance in melanin synthesis. As shown in Figure 1, the presence of THGP yielded signals of a complexed compound, whereas TMGP did not yield any signals of the complex. These findings show that THGP can interact with L-DOPA through dehydration between the two molecules. A previous crystal structure analysis revealed that THGP forms a complex with noradrenaline, which contains the same catecholamines as L-DOPA, by dehydration condensation via the diol group on catechol. During the formation of a complex of THGP with diol, the propanoic group originating from THGP forms a lactone ring and a coordinate bond with the vacant d-orbital of germanium, resulting in the formation of a pentacoordinate germanium (V) atom. Moreover, we also confirmed that THGP does not form a complex with tyrosine (phenol derivative of L-DOPA). Therefore, these data indicate that the interaction between THGP and L-DOPA requires both Ge-OH and vicinal diol and might be achieved via dehydration between the two molecules.

Therefore, we subsequently performed a study using mushroom tyrosinase and found that the synthesis of melanin from L-DOPA was inhibited by the addition of THGP. Additionally, THGP inhibited the tyrosinase-mediated reaction that produced melanin, and this inhibitory effect was not observed with TMGP (Figure 2). This result suggests that the inhibition of melanin synthesis by THGP was caused by complex formation with L-DOPA. Subsequently, the combination of THGP with kojic acid, which is reportedly a tyrosinase activity inhibitor, exerted a synergistic effect (Figure 4). It can be inferred that THGP and kojic acid suppress melanin synthesis via different mechanisms of action because THGP acts on the substrate as described above and kojic acid acts on the enzyme. In addition to kojic acid, α-arbutin, licorice flavonoid, and similar substances are used as whitening components [28,29,33]; therefore, these substances might exert similar synergistic effects with THGP against melanogenesis. Because these substances act on enzymes such as *tyrosinase*, *TRP-1*, and *TRP-2*, the inhibition of melanin synthesis can likely be enhanced by the combined use of THGP and these substances. These whitening agents have some known problematic side effects. Specifically, it has been reported that rhododenol caused vitiligo in Japan in 2014 and that rhododenol caused white spot damage on the skin by inhibiting enzymes involved in melanin production, such as tyrosinase, and inducing cell damage [34]. Hydroquinone has been used in the field of medicine and cosmetics due to its whitening effect, but its use is prohibited in many European countries due to its carcinogenic activity [35,36,37]. Therefore, the development of a safe cosmetic product with a whitening effect is needed [38]. This study demonstrates that THGP does not influence tyrosinase activity or the expression of genes involved in melanogenesis (Figure 5). In addition, previous studies have demonstrated that THGP exerts antitumor and antioxidant effects; thus, the side effects of THGP are likely limited [39,40,41,42]. Furthermore, because THGP did not affect cell proliferation at concentrations up to 5000 μM, THGP is considered safe up to this concentration (Figure 3).

In the cellular experiment, the extracellular melanin content was effectively decreased by the addition of THGP, but the amount of melanin that accumulated in the THGP-treated cells was greater than the control cells. It can therefore be considered that THGP has the effect of accumulating melanin in cells and suppressing the release of melanosome in addition to the inhibitory effect of melanin production by complex formation with L-DOPA. The mechanism, however, is not clear. Melanosomes mature in melanocytes and are released extracellularly, but THGP may inhibit melanosome maturation or melanosome release. It is necessary to clarify this mechanism in the future. However, as THGP did not affect the tyrosinase activity or the expression of melanin production-related genes, it may be involved in the melanin release from melanocytes and delivery to keratinocytes. In skin pigmentation, melanocytes are present in the spinous layer and deliver melanin to keratinocytes. Then, keratinocytes containing the melanin accepted from melanocytes progressively turn over in the epidermis. As a result, spots and darkening appear on the skin [43]. THGP not only suppresses melanin production but may also induce a whitening effect by inhibiting melanin transport. The inhibitory effects of the single use of THGP are shown in Figure 2, Figure 4, and Figure 6, and the inhibitory effects against the enzymatic reaction shown in Figure 2 are lower than those found in the other cell-based studies shown in Figure 4 and Figure 6. We believe that this difference originates from the suppressive effect of THGP on the release of melanin from melanocytes. Further investigation into this transport mechanism induced by the addition of THGP is strongly recommended. Moreover, as shown in Figure 6B, an additional study of L-DOPA revealed a mechanistic difference between THGP and kojic acid. The results of this study suggest that THGP is a novel substrate that suppresses the production of melanin and can potentially be applied in the cosmetic field.

Figure 7 shows a schematic of the effect of THGP on melanin production. In this study, we demonstrated that THGP forms a complex with L-DOPA by ^1^H-NMR. Furthermore, a test using mushroom tyrosinase proved that THGP exerts an inhibitory effect on melanin synthesis from L-DOPA. In addition, TMGP, which has a methyl group in place of the hydroxyl group present in THGP, did not exert an inhibitory effect on melanin synthesis, and this result suggests that THGP inhibits melanogenesis via complex formation with *cis*-diol structures. In an experiment using B16 4A5 melanoma cells, THGP also inhibited melanin synthesis. Furthermore, THGP did not affect tyrosinase activity or the expression of melanin production-related factors. These results suggest that THGP affects substrates rather than enzymes involved in melanin metabolism. Finally, kojic acid enhanced the inhibition of melanogenesis when administered in combination with THGP. In this study, tyrosinase enzyme and mouse melanoma cells were used, and similar results thus may not be obtained with human cells. Therefore, the inhibitory effects on cytotoxicity and melanin production need to be assessed in future studies using normal human melanocytes.

## 4. Materials and Methods

### 4.1. Reagents

The THGP used in this study was an aqueous hydrolysis solution of poly-*trans*-[(2-carboxyethylgerma) sesquioxane] synthesized at the Hakodate manufacturing plant of Asai Germanium Research Institute Co., Ltd. (Kawasaki, Japan). TMGP was synthesized at Asai Germanium Research Institute Co., Ltd. THGP was dissolved in water to a concentration of 500 mM, and the pH was adjusted to 7.05. This stock was diluted 100-fold to the working concentration of 5 mM using media.

### 4.2. Cell Culture

B16 4A5 cells (Riken BRC, Ibaraki, Japan), which are a mouse-derived melanoma cell line [30], were obtained from Riken Cell Bank and cultured in Dulbecco’s modified Eagle medium (DMEM) (Nissui Pharmaceutical Co., Ltd., Tokyo, Japan) supplemented with 10% fetal bovine serum (FBS) at 37 °C in 5% CO_2_. Once the cultures reached subconfluence, the cells were harvested with 0.01% trypsin and 1 mM ethylenediaminetetraacetic acid (EDTA) and cultured every 3–4 days at a split ratio of 1:4.

### 4.3. ^1^H-NMR

The ^1^H-NMR spectrum of each sample was measured using a Mercury Plus 300 MHz instrument (Agilent Technologies Inc., Santa Clara, CA, USA). The concentration and pH of each sample were adjusted to 5 mM and 7.05, respectively, and the samples were mixed at a molar ratio of 1:1. All solvents were prepared using heavy water. The light water (HOD: 4.80 ppm) remaining in the heavy water solvent was used as the standard for the assessment of chemical shifts. The measurement temperature was 25 °C, and the number of integrations was 16. Deuterium oxide (Kanto Chemical Co., Inc., Tokyo, Japan) and 40% sodium deuteroxide (NaOD; D, 99.5%) (FUJIFILM Wako Pure Chemical Corporation, Osaka, Japan) were used for the NMR measurements.

### 4.4. Enzymatic Reaction with Mushroom Tyrosinase

L-DOPA (4 mM, Tokyo Chemical Industry Co., Ltd., Tokyo, Japan) and tyrosinase (20 units, Sigma-Aldrich Corp., St. Louis, MO, USA) were reacted at 37 °C for 20 min with THGP or TMGP in a 96-well plate. In the combined test, 0 to 5000 μM THGP and 0 to 1000 μM kojic acid (FUJIFILM Wako Pure Chemical Corporation, Osaka, Japan) were added, and the absorbance at 405 nm was measured using an ARVO X3 (PerkinElmer Inc., Waltham, MA, USA).

### 4.5. MTS Assay

The cells were seeded at 5 × 10^3^ cells/well in a 96-well plate. After 24 h, 0 to 5000 μM THGP or 0 to 500 μM kojic acid was added. The cells were treated for 48 h, and an MTS assay was then performed using a CellTiter 96^®^ AQueous One Solution Cell Proliferation Assay kit (Promega Corp., Madison, WI, USA) according to the manufacturer’s recommended protocol. The absorbance at 495 nm was measured, and the results are presented as percentages relative to the nontreated group (designated 100%).

### 4.6. Measurement of the Melanin Content in Cells

The cells were seeded at 1 × 10^5^ cells/well on a 24-well plate, and after 24 h of incubation, THGP and/or kojic acid were added. In the combination tests, 0 to 500 μM THGP and 0 to 50 μM kojic acid were added. After 72 h of treatment, the supernatant was collected in 96-well plates, and the cells were dissolved by incubation at 60 °C for 20 min with 1 M NaOH (FUJIFILM Wako Pure Chemical Corporation). The absorbances of the cell lysate and supernatant were then measured at 405 nm. The results are presented as percentages relative to the nontreated group (designated 100%).

### 4.7. Measurement of Tyrosinase Activity

The cells were plated at a density of 1 × 10^6^ cells/well in a 60 mm dish. After 24 h, 0 to 5000 μM THGP was added, and after 48 h of treatment, the proteins were extracted from the cells using 0.1% Triton-X (Nacalai Tesque, Inc., Kyoto, Japan) in PBS (−). The cell lysate was frozen at −80 °C for 30 min and then thawed on ice. After centrifugation at 20,000× *g* for 20 min, the supernatant was collected. A Bradford assay (Bio-Rad Laboratories Inc., Hercules, CA, USA) was performed to quantify the protein concentration. The extracted protein sample (8 μg), including the enzymatic solution and 5 mM L-DOPA, was mixed and measured after incubation at 37 °C for 3 h. The absorbance at 405 nm was measured, and the results are presented as percentages relative to the nontreated group (designated 100%).

### 4.8. Quantitative Polymerase Chain Reaction (PCR)

The cells were seeded at 2 × 10^5^ cells/well in a six-well plate, and after 24 h of incubation, THGP (0–5000 μM) with or without 100 μM L-DOPA was added. After at least 24 h of treatment, total RNA was extracted using Isogen (Nippon Gene, Inc., Tokyo, Japan) according to the manufacturer’s recommended protocol. The extracted RNA was reverse transcribed by SuperScript III (Invitrogen, Life Technologies Corp., Carlsbad, CA, USA) with 1 μg of template. The PCRs consisted of 40 cycles of 95 °C for 5 sec and 60 °C for 30 sec and were performed using TB Green Premix Ex Taq II (Tli RNaseH Plus) (Takara Bio, Inc., Otsu, Japan) and a LightCycler 96 (Roche Diagnostics GmbH, Mannheim, Germany). The primers used in the PCR analysis are shown in Table 1, and β-actin was used as an internal control.

### 4.9. Measurement of the Melanin Content in Cells Treated with α-MSH

The cells were seeded at 1 × 10^5^ cells/well in a 24-well plate. After 24 h, 1 μM α-MSH (Peptide Institute Inc., Osaka, Japan) was added, and after 24 h, 0 to 5000 μM THGP and 0 to 500 μM kojic acid were added. After 72 h of treatment, the extracellular melanin contents were measured using the above-described measurement method (Section 4.6). The results are presented as percentages relative to the α-MSH group (designated 100%).

### 4.10. Measurement of the Melanin Content in Cells Treated With L-DOPA

The cells were seeded at 1 × 10^5^ cells/well in a 24-well plate. After 24 h, 0 to 5000 μM THGP, 0 to 500 μM kojic acid, and 250 μM L-DOPA were added, and after 72 h of treatment, the extracellular melanin contents were measured using the above-described measurement method (Section 4.6). The results are presented as percentages relative to the L-DOPA group (designated 100%).

### 4.11. Statistical Analysis

Dunnett’s method was used for all evaluations.

## Figures and Tables

**Figure 1 ijms-20-04785-f001:**
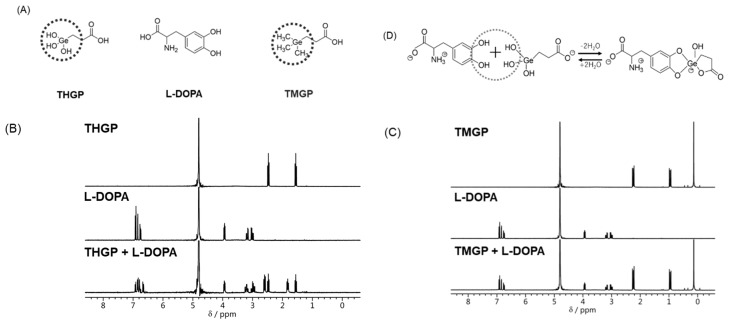
Formation of a complex of L-3,4-dihydroxyphenylalanine (L-DOPA) and 3-(trihydroxygermyl)propanoic acid (THGP). (**A**) Structures of THGP, L-DOPA, and 3-trimethylgermylpropanoic (TMGP). (**B**,**C**) The formation of a complex between THGP or TMGP and L-DOPA was analyzed by ^1^H-NMR (300 MHz). (**D**) Reaction for the formation of the THGP and-L-DOPA complex.

**Figure 2 ijms-20-04785-f002:**
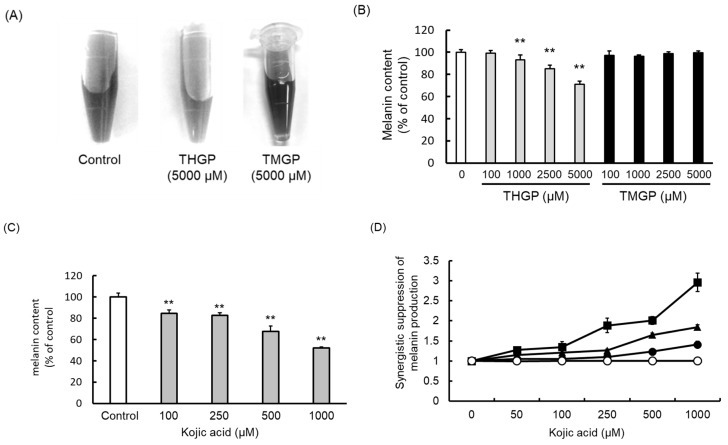
Inhibitory effect of THGP, TMGP, and kojic acid on the synthesis of melanin via a reaction mediated by mushroom tyrosinase. (**A**–**C**) THGP, TMGP, or kojic acid was mixed with L-DOPA and mushroom-derived tyrosinase and allowed to react at 37 °C for 20 min to synthesize melanin. (**D**) The suppressive effect of the combination of THGP and kojic acid on melanin synthesis was examined. The results were statistically analyzed using Dunnett’s method, and 0 µM THGP was used as the control. Circles: 0 µM THGP; filled circles: 1000 µM THGP; triangles: 2500 µM THGP; squares: 5000 µM THGP. The data are presented as the means ± SDs from six independent experiments. ** *p* < 0.01 vs. the control.

**Figure 3 ijms-20-04785-f003:**
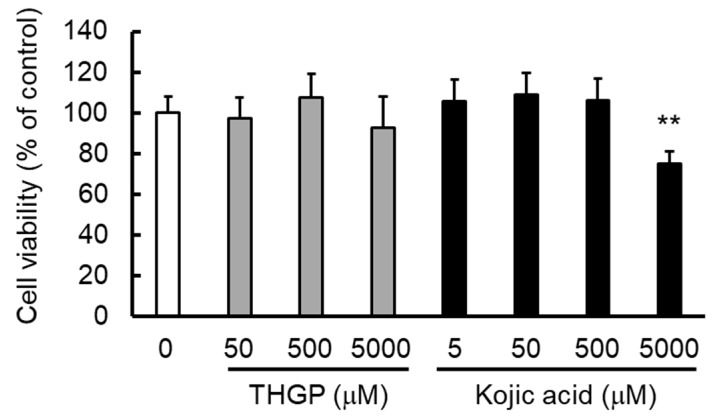
Cell viability after treatment with THGP or kojic acid. B16 4A5 melanoma cells were treated with THGP or kojic acid for 72 h, and the resulting cell viability was assessed by the MTS assay. The results were statistically analyzed by Dunnett’s method, and 0 μM THGP was used as the control. The results are presented as the means ± SDs from eight independent experiments. ** *p* < 0.01 vs. the control.

**Figure 4 ijms-20-04785-f004:**
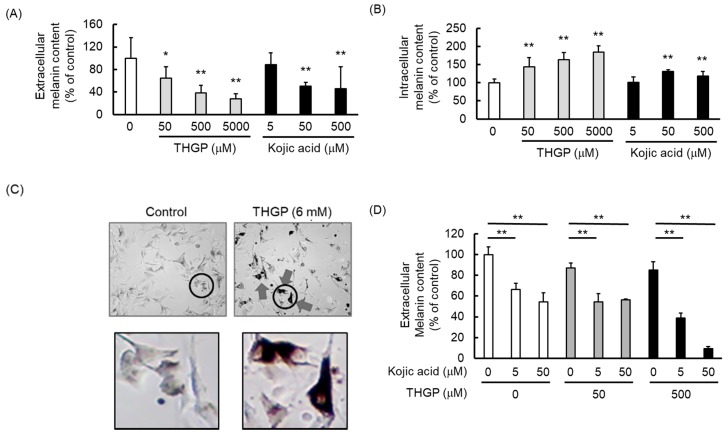
Inhibitory effect of THGP and kojic acid on melanogenesis in B16 4A5 melanoma cells. B16 4A5 cells were treated with THGP or kojic acid for 72 h, and the (**A**) extracellular and (**B**) intracellular melanin amounts were then evaluated. (**C**) Optical microscopy images of cells treated with THGP (6000 μM). The arrow means melanin accumulated in large amount in B16 4A5 and the circle means the enlarged location in the figure below. (**D**) The suppressive effect of the combination of THGP and kojic acid on melanin synthesis was evaluated. The results were statistically analyzed using Dunnett’s method, and 0 μM THGP served as the control. The data are presented as the means ± SDs from six independent experiments. * *p* < 0.05 and ** *p* < 0.01 vs. the control.

**Figure 5 ijms-20-04785-f005:**
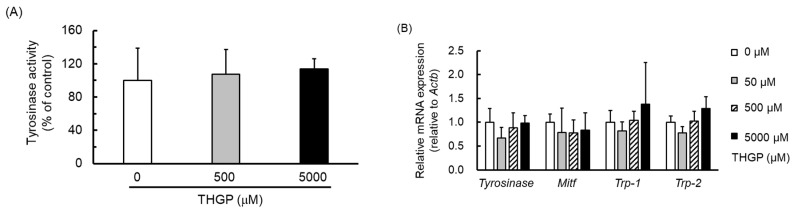
Effect of THGP on tyrosinase activity and the production of melanin production-related genes in B16 4A5 melanoma cells. (**A**) Tyrosinase activity was measured after treatment with THGP for 24 h. (**B**) The expression of genes after treatment with THGP for 24 h was assessed by real-time RT-PCR. (**C**) The expression of genes after treatment with THGP for 24 h with 100 μM L-DOPA was assessed by real-time RT-PCR. NC is nontreated L-DOPA and THGP. The values shown are relative to the expression of β-actin (Actb), a housekeeping gene. The results were statistically analyzed by Dunnett’s method, and 0 μM THGP with 100 μM L-DOPA served as the control. The data are presented as the means ± SDs from six independent experiments. ** *p* < 0.01 vs. the control.

**Figure 6 ijms-20-04785-f006:**
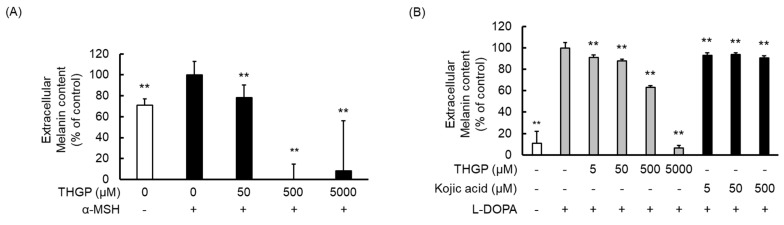
Inhibitory effect of THGP on melanin production after stimulation with α-MSH or L-DOPA. (**A**) The extracellular melanin levels in B16 4A5 melanoma cells under α-MSH activation were measured after treatment with THGP or kojic acid for 72 h. (**B**) The amount of extracellular melanin in B16 4A5 melanoma cells stimulated with L-DOPA was evaluated after treatment with THGP or kojic acid for 48 h. The results were statistically analyzed by Dunnett’s method, and 0 µM THGP with α-MSH or L-DOPA stimulation served as the control. “−” means untreated group and “+” means treated group about THGP, α-MSH or L-DOPA. The data are reported as the means ± SDs from six independent experiments. ** *p* < 0.01 vs. the control.

**Figure 7 ijms-20-04785-f007:**
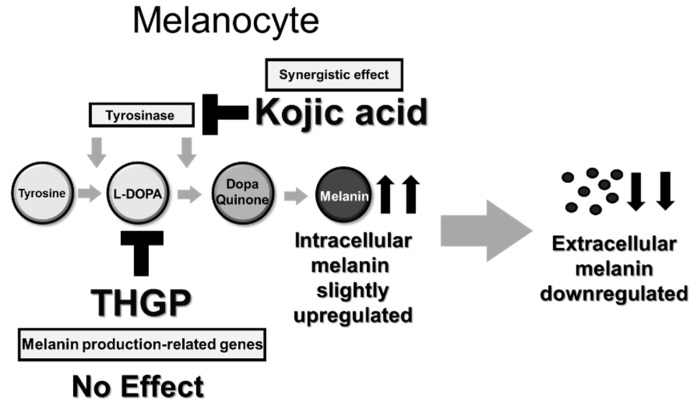
Mechanism underlying the inhibitory effect of the combination of THGP and kojic acid on melanin synthesis. Compared with the control condition, THGP administration inhibits the amount of extracellular melanin but slightly increases the amount of intracellular melanin. In addition, THGP does not affect tyrosinase activity or the expression of melanin production-related genes. Moreover, THGP shows a synergetic effect with kojic acid. “up arrow” means up-regulated, “down arrow” means down-regulated, “grey arrow” means reaction path about melanin syhthesis and “T arrow” means inhibitory effects.

**Table 1 ijms-20-04785-t001:** Primers used for real-time RT-PCR.

	Forward	Reverse
*Tyrosinase*	TTGCCACTTCATGTCATCATAGAATATT	TTTATCAAAGGTGTGACTGCTATACAAAT
*Trp-1*	ATGCGGTCTTTGACGAATGG	CGTTTTCCAACGGGAAGGT
*Trp-2*	CTCAGAGCTCGGGCTCAGTT	TGTTCAGCACGCCATCCA
*Mitf*	CGCCTGATCTGGTGAATCG	CCTGGCTGCAGTTCTCAAGAA
*Actb*	CTAAGGCCAACCGTGAAAAG	ACCAGAGGCATACAGGGACA

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
