# Peer review of "The Organogermanium Compound THGP Suppresses Melanin Synthesis via Complex Formation with L-DOPA on Mushroom Tyrosinase and in B16 4A5 Melanoma Cells"

_ijms, 2019, doi:10.3390/ijms20194785_

Round 1

Reviewer 1 Report

This manuscript is a study of the inhibitory melanin synthesis with 3-(trihydroxygermyl)propanoic acid (THGP). However, THGP did not increase tyrosinase activity, nor did it increase melanogenic related mRNA (Tyrosisase, Mitf, Trp-1,2) expression. This is a very interesting result.

In figure 4: In most of the case, intracellular and extracellular melanin content are proportional. However, the (A) and (B) results are shown in contrast in Figure 4. A detailed explanation about this is required.

The author said that THGP did not act as an enzyme, but inhibited melanin production by combining with L-Dopa. If the hypothesis is as above, the reliability of the hypothesis would be increased if the following experiments are added.

- Inducing melanin synthesis and mRNA analysis after adding of L-Dopa during cell

culture in media

- Data for addition of THGP after 24 hours cell culture with L-Dopa in media

Author Response

Thank you for your comment. Our responses to your various comments are shown below. We have revised the paper, and our changes based on the reviewer’s comments are shown in red in the revised manuscript.

Point 1

In figure 4: In most of the case, intracellular and extracellular melanin content are proportional. However, the (A) and (B) results are shown in contrast in Figure 4. A detailed explanation about this is required.

Response 1

We consider that THGP induces the accumulation of melanin in cells, suppresses the release of melanosome and exerts an inhibitory effect on melanin production by forming a complex with L-DOPA, but the mechanism is unclear. Melanosomes mature in melanocytes and are released extracellularly, but THGP might inhibit melanosome maturation or release. This mechanism needs to be clarified in the future.

Point 2

The author said that THGP did not act as an enzyme, but inhibited melanin production by combining with L-Dopa. If the hypothesis is as above, the reliability of the hypothesis would be increased if the following experiments are added.

- Inducing melanin synthesis and mRNA analysis after adding of L-Dopa during cell culture in media

- Data for addition of THGP after 24 hours cell culture with L-Dopa in media

Response 2

Twenty-four hours after the addition of 100 μM L-DOPA, the mRNA expression levels of various genes and the melanin amounts were measured (see the Results section).

The addition of L-DOPA changed the expression of genes involved in melanin synthesis, but THGP did not affect the expression of these genes compared to treated with 100 μM L-DOPA. This result suggests that THGP does not regulate gene expression and suppress melanin synthesis only by forming a complex with L-DOPA.

In addition, THGP also suppressed melanin synthesis during the investigated time course. This result was added to Fig. 5C.

Again, thank you for giving us the opportunity to strengthen our manuscript based on your valuable comments and questions. We have worked hard to incorporate your feedback and hope that these revisions persuade you to accept our submission.

Reviewer 2 Report

In the present manuscript, Azumi et al. investigated the functional role of THGP in melanogenesis in vitro and in vivo. The authors show that interaction of THGP and L-DOPA, and subsequently THGP inhibits melanin synthesis in vitro and mouse melanoma cells.

The manuscript by Azumi et al. is interesting and well-written. However, the authors should describe method section in detail.

Several figures are not easy to understand as well (Figure 1B and Figure 1C). I don’t understand how the authors measure melanin content in figure 1B and 1C in vitro experiment. Tyrosinase enzymatically oxidize L-DOPA to dopaquinone but not melanin, thus the authors should correct Y axis to L-DOPA oxidation or dopaquinone in figure 1B and 1C.

The authors should make sure THGP concentration which treated in cultured cells. I think 5 mM is too higher in cell culture condition. Thus, it is great if the authors describe how THGP stock solution was made and finally treated in cells.

I don’t understand Figure 1C, there are NO kojic acid non-treated group?

In Figure 6A, how melanin contents were normalized? The authors could describe how melanin contents were normalized (protein concentration or cell numbers) in this experiments.

Reference 45 should be removed.

Author Response

Thank you for your comment. Our responses to your various comments are shown below. We have revised the paper, and our changes based on the reviewer’s comments are shown in red in the revised manuscript.

Point 1

The manuscript by Azumi et al. is interesting and well-written. However, the authors should describe method section in detail.

Response 1

Some information was added to the Material and Methods section.

Point 2

Several figures are not easy to understand as well (Figure 1B and Figure 1C). I don’t understand how the authors measure melanin content in figure 1B and 1C in vitro experiment. Tyrosinase enzymatically oxidize L-DOPA to dopaquinone but not melanin, thus the authors should correct Y axis to L-DOPA oxidation or dopaquinone in figure 1B and 1C.

Response 2

As you noted, tyrosinase is an enzyme that converts L-DOPA to dopaquinone. However, dopaquinone is unstable and is thus rapidly oxidized to eventually form melanin. Therefore, because it is difficult to examine the amount of dopaquinone, the inhibition rate of the enzyme reaction was calculated by measuring the absorbance of 405 nm, which is the specific absorbance of melanin. The Y-axis in Fig. 2B shows the measured amounts of melanin, and we thus believe that the description is correct. I added this sentence to the Discussion section.

Point 3

The authors should make sure THGP concentration which treated in cultured cells. I think 5 mM is too higher in cell culture condition. Thus, it is great if the authors describe how THGP stock solution was made and finally treated in cells.

Response 3

THGP was dissolved in water to a concentration of 500 mM, and the pH was adjusted to 7.05. This stock was diluted 100-fold to the working concentration of 5 mM using media. The methods used to prepare and add the stock solution of THGP are described in the Materials and Methods section.

Point 4

I don’t understand Figure 1C, there are NO kojic acid non-treated group?

Response 4

The data obtained with kojic acid alone were added to Fig. 2C.

In Fig. 2D, the theoretical value for the combination of THGP and kojic acid is considered the standard (with a value of 1), and the synergistic effect of THGP was calculated in comparison to this standard. For example, 5000 μM THGP alone resulted in a melanin content of 67.8% compared with the control, and 1000 μM kojic acid alone yielded an inhibition rate of 51.9%. The theoretical value was found to equal 35.2%, which corresponds to the multiplication of 67.8% by 51.9%. However, because the experimentally measured inhibition rate was 12%, the synergistic effect was 2.93-fold higher than the theoretical value.

Point 5

In Figure 6A, how melanin contents were normalized? The authors could describe how melanin contents were normalized (protein concentration or cell numbers) in this experiment.

Response 5

In Fig. 6A, stimulation with α-MSH or L-DOPA was considered the control, and the resulting amount of melanin was set to 100%. A sentence providing this information was added to the Materials and Methods section.

Point 6

Reference 45 should be removed.

Response 6

Reference 45 was deleted.

Again, thank you for giving us the opportunity to strengthen our manuscript based on your valuable comments and questions. We have worked hard to incorporate your feedback and hope that these revisions persuade you to accept our submission.

Reviewer 3 Report

The paper by Azumi et al reports the effects of an organogermanium compound on melanin formation in vitro and on cellular models. The compound investigated has already been reported to possess the ability to complex a variety of compounds of biological relevance possessing a cis diol functionality, and to exert various biological effects with low cytotoxicity.

In this paper the Aus obtain evidence by NMR analysis for the formation of a complex between Dopa and the organogermanium compound and ascribe to this the reduced melanin formation observed in vitro by mushroom tyrosinase oxidation of DOPA in the presence of such compound.  On the other hand in melanoma cell lines lower levels of extracellular melanin is observed, but with concomitant pigment accumulation inside the cell.

These effects are apparently synergistically enhanced by koijc acid in a dose dependent manner.

The general experimental design of the work is acceptable, and the finding may be of interest, but there a number of critical issues that should be addressed before the work may be considered for publication.

General concern. Tyrosinase is known to possess two activities and can use as substrates either phenols, which undergo monooxygenation (monophenolase or cresolase activity), or catechols, which undergo oxidation (diphenolase or catecholase activity), leading in both cases to ortho-quinones. Although the view that phenol oxidation involves a hydroxylation step followed by catechol oxidation is still commonly claimed in the literature, in the monophenolase mechanism the ortho-quinone is formed directly from the phenol, and catechol formation is NOT an intermediate step [Ramsden, C.A.; Riley, P.A. Tyrosinase: the four oxidation states of the active site and their relevance to enzymatic activation, oxidation and inactivation. Bioorg. Med. Chem. 2014, 15, 2388–2395]. This implies that while in vitro the activities can be separately tested using a monophenol or a diphenol substrate, this is not possible in vivo where tyrosine is the melanogenic precursor and dopa is not formed as intermediate. Dopa accumulated in melanocytes is the result of a redox exchange between dopaquinone and leucodopachrome generated in the cyclization of dopaquinone.

Thus while in vitro inhibition of dopa complexation may be conceivable as an approach to decrease melanin formation this would in principle not possible in vivo where dopa is not a true intermediate in the melanin formation pathway.

Moreover,  the differences of activities of mushroom tyrosinase and mammalian tyrosinase are well known and are exemplified by substrate specificity. For example Kojic acid as inhibitor has an IC50 value of 60 microM for mushroom tyrosinase but is a very poor inhibitor (> 500 microM) of human tyrosinase. (Mann et al.  Inhibition of human tyrosinase requires molecular motifs distinctively different from mushroom tyrosinase. J. Invest. Dermatol. 2018, 138, 1601-1608; Solano et al. Hypopigmenting agents: an updated review on biological, chemical and clinical aspects Pigment Cell Res. 2006, 195, 550-571). This is true also for beta-arbutin. These considerations should be take into account when proposing a melanogenesis inhibitor based on Dopa complexation.

 In particular

Title: the formation of the complex of dopa with the organogermanium compound has been shown only in vitro so the title should reflect more correctly the results.

Introduction: line 57-60 is this the statement of the present work or whatelse? Please clarify

Results: section 2.1 dopa-organogermanium complex. Splitting of the resonances of dopa may be evidence of an interaction between the two compounds possibly partial complex formation but the site of the complexation on dopa can not be inferred if not using model compounds like in the case of THGP/TMGP (also the amino and carboxyl group of the aminoacid could act as ligands). Anyway,  if the o-diphenol moiety is primarily involved in the interaction one would expect that the aromatic signals undergo a larger shift compared to other signals. This issue should be checked and an inset showing a magnification of this region could be included and discussed. What is the evidence for the participation of the carboxyl group of the organogermanium to the complex?

Section 2.2: though the toxicity of organogermanium compound seems to be very low, the concentrations tested and those effective are indeed very high. An IC50 value should be provided.

lines 90-92 this conclusion is not correctly expressed , please address

Fig. 2C what is the concentration of kojic acid micromolar ?

Section 2.3 Which melanin is shown in figure 4d ?

The effects on intra and extra cellular melanin contents are indeed intriguing and no convincing interpretation is provided. In addition, for human tyrosinase kojic acid can not be regarded as an effective inhibitor at concs as high as 500 micromolar (as shown also in figure 5 and  6) , so the synergistic effects would be even more interesting if interpreted.

Discussion lines 192-194  what is the evidence for water elimination in complex formation?

The possible application of the organogermanium compound  as skin whitening agent should be critically evaluated in view of the large concentration needed and the much different effects observed with mushroom and human tyrosinase.

Author Response

Thank you for your general comments and advice. We would like to do the research recommended in your comments in the future.

Responses to your comments are provided below. We have revised the paper, and our changes based on the reviewer’s comments are shown in red in the revised manuscript.

Point 1

Title: the formation of the complex of dopa with the organogermanium compound has been shown only in vitro so the title should reflect more correctly the results.

Response 1

The title was changed to “The organogermanium compound THGP suppresses melanin synthesis via complex formation with L-DOPA on mushroom tyrosinase and in B16 4A5 melanoma cells”.

Point 2

Results: section 2.1 dopa-organogermanium complex. Splitting of the resonances of dopa may be evidence of an interaction between the two compounds possibly partial complex formation but the site of the complexation on dopa can not be inferred if not using model compounds like in the case of THGP/TMGP (also the amino and carboxyl group of the aminoacid could act as ligands). Anyway, if the o-diphenol moiety is primarily involved in the interaction one would expect that the aromatic signals undergo a larger shift compared to other signals. This issue should be checked and an inset showing a magnification of this region could be included and discussed. What is the evidence for the participation of the carboxyl group of the organogermanium to the complex?

Discussion lines 192-194 what is the evidence for water elimination in complex formation?

(Line 192-194: This finding shows that THGP can interact with L-DOPA by dehydration between the two molecules. Therefore, we next performed a study using mushroom tyrosinase, and the synthesis of melanin from L-DOPA was inhibited by the addition of THGP.)

Response 2

As demonstrated by previous crystal structure analyses, THGP forms a complex with noradrenaline, which contains the same catecholamines as L-DOPA, by dehydration condensation via diol on catechol. During formation of the complex of THGP with diol, the propanoic group originating from THGP forms a lactone ring and a coordinate bond with the vacant d-orbital of germanium, resulting in the formation of a pentacoordinate germanium (V) atom. Moreover, we have also confirmed that THGP does not form a complex with tyrosine (phenol derivative of L-DOPA). Therefore, these data indicate that the interaction between THGP and L-DOPA requires Ge-OH and vicinal diol and can interact via a dehydration reaction between the two molecules.

We have added new sentences providing the above-described explanation in the Discussion section.

Point 3

Section 2.2: though the toxicity of organogermanium compound seems to be very low, the concentrations tested and those effective are indeed very high. An IC50 value should be provided.

Response 3

We apologize for providing the incorrect unit of concentration in Fig. 2B. For melanin synthesis in B16 4A5 cells, the IC50 of THGP is approximately 230 μM, and the IC50 of kojic acid is 500 μM.

We changed Fig. 2B and added a relevant sentence in Subsection 2.3 in the Results section.

Point 4

lines 90-92 this conclusion is not correctly expressed, please address

(line 90-92: This result suggested that TMGP methyl groups cannot suppress melanin formation, while THGP hydroxyl groups suppress melanin formation. We hypothesized that THGP inhibits the synthesis of melanin by complexing with L-DOPA.)

Response 4

The text was modified as follows:

This result suggests that the hydroxyl group on THGP is important for melanin synthesis, and thus, THGP might inhibit the synthesis of melanin by complexing with L-DOPA via the hydroxyl group of THGP.

Point 5

Fig. 2C what is the concentration of kojic acid micromolar ?

Response 5

We modified Fig. 2D. The doses are 0, 50, 100, 250, 500, and 1000 μM.

Point 6

Section 2.3 Which melanin is shown in figure 4d ?

Response 6

The measurements of the amount of melanin include extracellular melanin. The term “extracellular” was added to the Y-axis of Figs. 4D, 6A, and 6B.

Point 7

The effects on intra and extra cellular melanin contents are indeed intriguing and no convincing interpretation is provided. In addition, for human tyrosinase kojic acid can not be regarded as an effective inhibitor at concs as high as 500 micromolar (as shown also in figure 5 and 6), so the synergistic effects would be even more interesting if interpreted.

Response 7

THGP can potentially induce the accumulation of melanin in cells, suppress the release of melanosome and exert an inhibitory effect on melanin production via complex formation with L-DOPA, but the mechanism is unclear. Melanosomes mature in melanocytes and are released extracellularly, but THGP might inhibit melanosome maturation or release. It is thus necessary to clarify this mechanism in the future. This information was added to the Discussion section.

Point 8

The possible application of the organogermanium compound as skin whitening agent should be critically evaluated in view of the large concentration needed and the much different effects observed with mushroom and human tyrosinase.

Response 8

In this experiment, tyrosinase enzyme and mouse melanoma cells were used, and similar results might not be obtained with human cells. Therefore, it is necessary to examine the inhibitory effects on cytotoxicity and melanin production in future studies using normal human melanocytes. This information was added to the Discussion section.

Again, thank you for giving us the opportunity to strengthen our manuscript based on your valuable comments and questions. We have worked hard to incorporate your feedback and hope that these revisions persuade you to accept our submission.

Round 2

Reviewer 3 Report

The aus duly addressed the issues raised.  Some critical issues have been clarified